# A Scalable Crawling Algorithm Utilizing Noisy Change-Indicating Signals

## ABSTRACT

Web refresh crawling is the problem of keeping a cache of web pages fresh, that is, having the most recent copy available when a page is requested, given a limited bandwidth available to the crawler. Under the assumption that the change and request events, resp., to each web page follow independent Poisson processes, the optimal scheduling policy was derived by Azar et al. [1]. In this paper, we study an extension of this problem where side information indicating content changes, such as various types of web pings, for example, signals from sitemaps, content delivery networks, etc., is available. Incorporating such side information into the crawling policy is challenging, because (i) the signals can be noisy with false positive events and with missing change events; and (ii) the crawler should achieve a fair performance over web pages regardless of the quality of the side information, which might differ from web page to web page. We propose a scalable crawling algorithm which (i) uses the noisy side information in an optimal way under mild assumptions; (ii) can be deployed without heavy centralized computation; (iii) is able to crawl web pages at a constant total rate without spikes in the total bandwidth usage over any time interval, and automatically adapt to the new optimal solution when the total bandwidth changes without centralized computation. Experiments clearly demonstrate the versatility of our approach.

## CCS CONCEPTS

• **World Wide Web**; • **Web searching and information discovery**; • **Web search engines**; • **Web crawling**;

## KEYWORDS

Search & Information retrieval, Web page indexing, Crawling policy

**ACM Reference Format:**
Anonymous Author(s). 2018. A Scalable Crawling Algorithm Utilizing Noisy Change-Indicating Signals. In *Proceedings of Make sure to enter the correct conference title from your rights confirmation emai (Conference acronym 'XX).* ACM, New York, NY, USA, 17 pages. https://doi.org/XXXXXXX.XXXXXXX

## 1 INTRODUCTION

Efficient web refresh crawling is one of the most fundamental data management problems in web search. Web pages are the simplest

and most ubiquitous source of information on the Internet, therefore, extracting and organizing their information content is a crucial task. The first step in this process is acquiring the information, which means that web pages are regularly crawled to ensure that the information at the search engine is up-to-date. However, the scale of the problem is enormous since, as of today, there are trillions of web pages available online, and hence efficient, resource-aware crawling algorithms are of great practical interest.

In this paper, we consider the problem of designing scheduling policies for refreshing web pages in a local cache with the objective of maximizing the probability that incoming requests to the cache are processed and served with the latest version of the page. An efficient web page caching policy, thus, aims to schedule refresh crawl events (i.e., crawling a web page) between subsequent content change and content request events in order to have a fresh copy upon request. The issue of course is that the scheduler needs to check if a web page has changed, hence web pages need to be polled regularly, which uses bandwidth, and polling them only provides partial information about their change process, such as a single bit indicating whether the web page has changed since it was last refreshed [4].

It has been estimated that up to 53% of the crawling traffic of major search engines is redundant [5] in the sense that the crawler visits an unchanged version of a web page. There is a growing awareness of the environmental and monetary cost of this redundancy, which motivated initiatives to provide additional signals for crawlers, allowing them in theory to improve their crawling policies. One form of this effort is for web servers to actively send so-called *web pings*, which notify interested parties about content changes, so that some active crawling traffic for detecting changes can be saved. There are different types of web pings, such as signals from sitemaps, content delivery networks, web servers, etc., which can all provide content change notifications to downstream services, such as web crawlers or proxy servers. Kolobov et al. [7] recognized the importance of side information and proposed an algorithm under the assumption of complete and accurate change information for URLs (i.e., web pages; throughout the paper we use the expressions URLs and web pages interchangeably) with side information. We provide empirical evidence and conceptual arguments that this assumption is too strong and extend the model to cover noisy signals as well.

In the classical crawling setup, the change and request sequences are assumed to obey certain stochastic processes [3], which allows to derive policies with some optimality guarantee, even if the change and request sequences are not fully observable. The most standard assumption is that the sequence of changes and requests, resp., to each web page are independent Poisson processes [3, 4, 12]. For this setting, Cho and Garcia-Molina [3] derived a convex optimization problem to find the optimal refresh rates for the web pages if the arrival rates of the Poisson processes are known. Azar

et al. [1] presented an explicit efficient algorithm to find the solution, which, in fact, crawls every web page at a fixed, page-specific rate. However, this results in a non-uniform total crawling rate, which is problematic in practice (as the resources available to the crawler have to match the peak rate). Therefore, they also proposed a method to transfer a variable-rate continuous-time (*continuous* for short) crawling policy with page-specific fixed crawl rates to a policy with constant total crawl rate; such a policy schedules crawl events at regular intervals, that is, at discrete time steps (hence, such policies will be referred to as *discrete* policies). This reduction relies crucially on crawling all pages at a fixed rate. In the presence of side information, the optimal continuous policy does not satisfy fixed rates and prior work has not provided reductions to discrete policies for this setup [7].

An alternative method to derive discrete policies from continuous ones based on Lagrange multipliers has been proposed in a Google blog post [10]. A benefit of their reduction is that one can modify problem parameters such as the change rate of a URL or its importance on-the-fly without requiring a global recomputation of the solution. This makes this method especially appealing for real-world scenarios where such parameters are continuously estimated [4, 11].[1]

Another line of research leverages ideas from online learning and reinforcement learning to learn a crawling policy beyond the Poisson model [6, 7]; however, reinforcement learning methods are computationally infeasible in large-scale environments.

In this paper we build on the Poisson model and the continuous to discrete reduction via Lagrange-multipliers and make the following contributions:

(1) Motivated by the empirical evaluation of information available to real-world crawlers, we extend the crawling problem with imperfect (i.e., *noisy*) *change-indicating signals* (*CI signals* or *CISs*, for short).We derive the optimal (continuous) crawling strategy for this model under a global bandwidth constraint.

(2) Based on the above policy, we also derive a discrete crawling strategy that is able to use CISs. This strategy is practically appealing for several reasons: (i) up to a final arg max operation, it is fully decentralized and parallelizable over web pages both in computation and memory; (ii) the total crawl rate is constant over time without spikes in bandwidth usage over any time interval; (iii) the method automatically adapts to changes of the bandwidth constraint without centralized computation; and (iv) regular changes to the estimations of model parameters do not induce any additional computational overhead.

(3) We provide an extensive empirical study showing a clear benefit of our method on semi-synthetic data.

The paper is organized as follows: In Section 2 we provide empirical evaluation of CIS justifying the need of our extended model. In Section 3, we introduce the crawling setup, its objective function and the formal definition of CI signals. In Section 4, we recall the

optimization view of the crawling problem which allows to compute optimal continuous policies, and introduce a principled way to incorporate change-indicating signals into this setting. We introduce a general approach to derive easy-to-implement and scalable greedy polices based on continuous ones in Section 5. Experimental studies are presented in Section 6, followed by our conclusions and future plans in Section 7. All the proofs of our theoretical results are relegated to Appendix A, and additional experiments are presented in Appendices B, D and C.

## 2 CHANGE-INDICATING SIGNALS

In this section, we present some to basic statistics about change detection signals in general and in the dataset we use in our experiments.

We take the dataset from [7], published in 2019, as a starting point. It contains 18,532,314 URLs which were crawled intensively over a period of 2 weeks to compute their empirical change rates. Additionally, they recorded for which sites they obtained side-information and the importance of the pages according to the Bing production crawler based on PageRank and popularity.

While only 4% of the URLs has side information, they represent 26.4% of importance-adjusted weight, which is why the question of including side information is relevant even when the adoption is not wide-spread yet. In their work, Kolobov et al. [7] assume that these CISs are noiseless, that is, for every change there is a CIS generated, and every CIS corresponds to a valid change.

Concerning the quality of CISs, it is worth noting that defining a change itself is a complex question, since what counts as a change depends on the actual use case: For example, should correcting a typo on a web page count as a change, or changing the header, or some on-the-fly generated content, such as an ad? Because of this, there is no unified definition of what counts as change; Kolobov et al. [7] use the non-public change definition of their (Bing) production crawler, we also use our own definition, and CIS providers use their own, as well. As a result, CISs are necessarily noisy from the viewpoint of their users (even if they are perfect according to the change definition of their providers), and hence the model of complete and error free signals, used in [7], is too simplistic to capture the real world.

In fact, we measured the quality of the sitemap signals for the web pages in their dataset with declared (perfect) sitemaps and found that the *precision* of these signals (i.e., the proportion of time there is a change following a signal) is below 0.2, and their *recall* (i.e., the proportion of time when a change is accompanied with a CIS) is below 0.5, hence they are far from perfect. To give more insights into the quality of CISs, we measured the distribution of precision and recall of all the web pages we have sitemap signals for, and the resulting importance-weighted histograms (using our own confidential definition of importance, which is also based on PageRank and should be similar to that of Bing) are shown in Figure 1. These results clearly demonstrate that sitemap signals are noisy, but they are in general better than what we found for the URLs in the dataset of [7]. Importantly, there are very few pages which have perfect signal with precision and recall that are higher than 0.8.

---

[1]In this work we do not consider the estimation of the standard parameters of the Poisson model (i.e., the change rate), but discuss the estimation of the parameters describing the behavior of the CI signals in Appendix E.

 

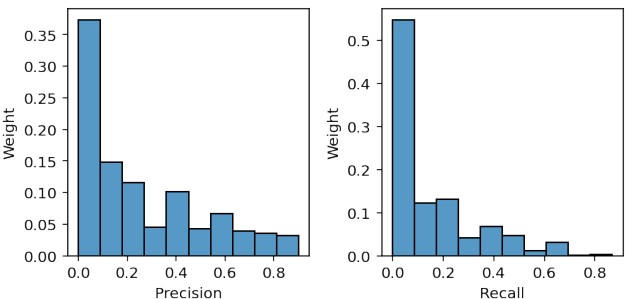

**Figure 1: Histogram of precision and recall for URLs with sitemap signals. The histogram is computed by weighting the pages with their importance.**

## 3 SETUP

*Classical crawling model.* We are given $m$ web pages to be cached that are indexed by unique URLs. For the sake of presentation, we index them by $[m] = \{1, \dots, m\}$ and we consider the set of URLs to be fixed. The classical caching problem for a web page $i$ is modelled by three processes: 1) the request process; 2) the change process; and 3) the refresh (or crawling) process. The request and change processes are assumed to be independent Poisson processes with parameters $\mu_i$ and $\Delta_i$, respectively. We also assume that these Poisson processes are independent across all the web pages. In a practical system, the request processes can be fully observed, while the change process of a web page can only be partially observed by comparing its last cached version to the current one when a crawl event is triggered. The goal of caching is to have a fresh copy of the web page when a request event happens. So a good crawling policy tries to refresh each web page between every subsequent change and request events. Without further constraints, the naive solution is to continuously refresh each web page all the time, which is obviously not an implementable policy in practice (due to limitations in computations and communications). To obtain a more realistic setting, we assume a *bandwidth constraint*, which either limits the overall frequency of crawl events in expectation, or sets a hard constraint on the minimal time interval between two subsequent crawl events.

*Policies.* A crawling policy decides at any time $t \in [0, \infty)$ (potentially based on the history up to time $t$) whether it crawls web pages, thereby over time generating an infinite sequence of crawl events: crawl events are specified by time-web page pairs where $(t, i)$ defines a crawl event for web page $i$ at time $t$, and the sequence of crawl events is denoted as $\mathcal{T} = ((t_1^{\mathrm{CR}}, i_1), (t_2^{\mathrm{CR}}, i_2), \dots)$, where $t_s^{\mathrm{CR}} \le t_{s+1}^{\mathrm{CR}}$ for any $s \in \mathbb{N}$. We denote the crawl-event sequence of an individual web page $i$ by $\mathcal{T}_i = (t_{i,1}^{\mathrm{CR}}, t_{i,2}^{\mathrm{CR}}, \dots)$, where $t \in \mathcal{T}_i$ if and only if $(t, i) \in \mathcal{T}$. As mentioned before, we consider two policy classes under bandwidth-budget constraints:

- **Continuous:** the average number of crawl events over time is asymptotically upper bounded by a budget $R$ almost surely (a.s.): $\limsup_{T \to \infty} \frac{1}{T} \max\{i \in \mathbb{N} \mid t_i \le T\} \le R$ almost surely.
- **Discrete:** the crawl events are uniformly triggered at fixed times: $t_j^{\mathrm{CR}} = \frac{j}{R}$.

The policies in the continuous class specify the time of each crawl event, while the policies in the discrete class choose a web page to crawl at each time step $t_j^{\mathrm{CR}}$ by treating the time when a crawl event can happen as a constraint. Furthermore, an optimal solution of the continuous class only needs to satisfy the total bandwidth constraint asymptotically, while an optimal solution of the discrete class satisfies the total bandwidth constraint over any time interval. The infrastructure and bandwidth constraints as well as the volatile web environment make the continuous class of policies less practical. We have assumed a uniform, fixed rate for the discrete class of policies. However, our method can be easily applied to other discrete classes with non-uniform rates (e.g. when the total bandwidth changes). Optimizing over the discrete class is a hard combinatorial problem and it is significantly easier to find the optimal solution in the continuous class. Prior work followed the recipe of first computing the optimal solution in the continuous case and then approximating the continuous crawl rate in the discrete policy class [1]; in this paper we also take this approach.

*CI signals.* We extend the classical model described above by incorporating CI signals as additional observations. We assume that for each web page $i$, for every change a CI signal[2] is available with probability $\lambda_i$ for some $\lambda_i \in [0, 1]$, (independently for each change). Given this assumption, the (Poisson) change process for web page $i$ can be split into two independent Poisson processes, a signalled change process with rate $\lambda_i \Delta_i$ and an unsignalled – and hence also directly unobserved – change process with rate $(1 - \lambda_i)\Delta_i$. Additionally, we also allow the presence of false CI signals, and assume that these signals are generated by independent Poisson processes with rate $\nu_i$ for each web page $i$. The decision maker cannot distinguish whether a CI signal was produced by the signalled change process or the false signal process. In summary, for each web page $i$, one observes an additional sequence of CI signals $C_i = (t_{i,1}^{\mathrm{CIS}}, t_{i,2}^{\mathrm{CIS}}, \dots)$ generated by a Poisson process with rate $\gamma_i = \lambda_i \Delta_i + \nu_i$, which contains both true and false CI signals. We denote the rate of the unobserved change process by $\alpha_i = (1 - \lambda_i)\Delta_i$.[3] The quality of CISs for page $i$ is often measured by their precision and recall: *precision*, the probability that a CI signal corresponds to a real change, is given by $\lambda_i \Delta_i / \gamma_i$, while *recall*, the the probability that a change is indicated by a signal, is equal to $\lambda_i$ by definition.

*Objective.* The objective of the crawl scheduler is to maximize the expected number of requests that are served with a fresh copy of the corresponding web page, defined, for a policy $\pi$, as[4]

$$O(\pi) := \liminf_{N \to \infty} \frac{1}{N} \sum_{n=1}^{N} \mathbb{E}_\pi \left[ \mathbb{I}_{\left\{ \text{the } n\text{-th request hits a fresh copy} \right\}} \right] .$$

In the notation we explicitly included the dependence on $\pi$ to the expectation; note, however, that the expectation is also taken over the randomness induced by the environment, in particular by the different Poisson processes. This dependence is also omitted later.

---

[2]It is straightforward to extend the model to multiple independent sources of CI signals. We consider a single signal for the sake of presentation.

[3]Kolobov et al. [7] study the regime where $\nu_i = 0$ (noiseless) and $\lambda_i \in \{0, 1\}$ (complete or no information at all).

[4]While we define limiting quantities in terms of lim inf and lim sup for generality, they can be replaced with lim for the optimal policies we derive, as the corresponding limits exist.

To simplify this objective, we recall two important properties of Poisson processes:

- **Uniformity**: Events that obey a Poisson process will be uniformly distributed over any time interval.
- **Merging**: A collection of independent Poisson processes with rates $(\mu_1, \ldots, \mu_m)$ is equivalent to a single Poisson process with rate $\mu = \sum_i \mu_i$ where each time an event is generated, it is assigned a random index from $[m]$ drawn from a discrete distribution with parameters $(\mu_1/\mu, \ldots, \mu_m/\mu)$.

We assume the request events obey Poisson processes, therefore due to uniformity and merging, we can rewrite the objective as

$$O(\pi) = \sum_{i=1}^{m} \liminf_{T \to \infty} \frac{\tilde{\mu}_i}{T} \int_0^T \mathbb{P}_\pi \left[ E_i(t) \mid \mathcal{H}_t \right] \, dt \,,$$

where $E_i(t)$ denotes the event that page $i$ is fresh at time $t$, $\tilde{\mu}_i = \mu_i/(\sum_{j=1}^m \mu_j)$ is the normalized importance and $\mathcal{H}_t = (\mathcal{T}_i \cap [0,t), C_i \cap [0,t))_{1,\ldots,m}$ is the conditioning on all observable events before time $t$. Let

$$\tau_i^{\text{ELAP}}(t) = t - \max\{t_{i,k}^{\text{CR}} \mid k \in \mathbb{N}, \, t_{i,k}^{\text{CR}} \leq t\}$$

be the elapsed time since the last crawl for web page $i$ at time $t$ and

$$n_i^{\text{CIS}}(t) = \left| C_i \cap (t - \tau_i^{\text{ELAP}}, t] \right|$$

be the number of CI signals received since the last crawl event of web page $i$. Then the conditional probability of page $i$ being fresh is

$$\mathbb{P}\left[ E_i(t) \mid \mathcal{H}_t \right] = \exp(-\alpha_i \tau_i^{\text{ELAP}}(t)) \cdot \left( \frac{v_i}{\gamma_i} \right)^{n_i^{\text{CIS}}(t)} . \quad (1)$$

The first factor is the probability that a Poisson process with rate $\alpha_i$ did not produce any event in an interval of length $\tau_i^{\text{ELAP}}(t)$. The second factor follows from the fact that all CI signals are i.i.d. events and with every event, the probability that the CI signal was a false positive is $\frac{v_i}{\gamma_i}$. We can reduce the state (probability of freshness) to a single number by observing that the freshness reduction of each CI signal is equivalent to that of not observing any signal for a certain time. Motivated by this fact, let

$$\beta_i = -\frac{\log(v_i/\gamma_i)}{\alpha_i} \quad \text{and} \quad \tau_i^{\text{EFF}}(t) = \tau_i^{\text{ELAP}}(t) + \beta_i n_i^{\text{CIS}}(t)$$

be the time-equivalent of a single CI signal and the effective elapsed time, respectively. Then (1) equals to $\exp(-\alpha_i \tau_i^{\text{EFF}}(t))$, and the objective becomes

$$O(\pi) = \sum_{i=1}^{m} \liminf_{T \to \infty} \tilde{\mu}_i \mathbb{E}_{\pi, t \sim \text{unif}([0,T])} \left[ \exp(-\alpha_i \tau_i^{\text{EFF}}(t)) \mid \mathcal{H}_t \right] . \quad (2)$$

## 4 CONTINUOUS POLICIES

In this section, we focus on how to find optimal continuous policies which optimize (2) under some bandwidth constraint. Our derivation can be viewed as a generalization of the approach of Azar et al. [1], but here we make use of noisy CISs. All proofs are deferred to Appendix A. First, we narrow down the policy class of potentially optimal policies.

LEMMA 1. *Any policy optimizing the objective $O(\pi)$ defined in (2) has a decision rule such that it triggers a crawl event $(\tau, i)$ when $\tau_{i,\tau}^{\text{EFF}} \geq \iota_i$ for some fixed threshold vector $\boldsymbol{\iota} = (\iota_1, \ldots, \iota_m) \in (0, \infty]^m$.*

From now on, let $\pi(\boldsymbol{\iota})$ denote the thresholded policy which crawls web page $i$ when $\tau_{i,\tau}^{\text{EFF}} \geq \iota_i$. The class of such thresholded policies contains at least one optimal policy by Lemma 1, so we restrict our attention to this class. Also, because of Lemma 1, any thresholded policy is *separable* over web pages, which means that the threshold $\iota_i$ determines the performance of the policy $\pi(\boldsymbol{\iota})$ on web page $i$ regardless what thresholds are picked for the rest of the web pages. Nonetheless, solving the optimization problem (2) under bandwidth constraint is still not straightforward, since the length of the crawl intervals are random quantities and so are the crawling frequencies. Motivated by this observation, we now focus only on a single page: the crawl frequency of policy $\pi(\boldsymbol{\iota})$ for web page $i$ and parameters $\mathcal{E}_i = (\alpha_i, \beta_i, \gamma_i, \tilde{\mu}_i)$ can be computed as

$$f(\iota_i, \mathcal{E}_i) := \limsup_{T \to \infty} \frac{1}{T} \mathbb{E}_{\pi(\boldsymbol{\iota})} \left[ |\mathcal{T}_i \cap [0,T]| \right] .$$

Note that the crawl frequency for web page $i$ depends only on $\iota_i, \mathcal{E}_i$, and no other parameters. Also note that the number of crawl events for an interval is a random quantity even for a thresholded policy $\pi(\boldsymbol{\iota})$, since the sequence $\tau_{i,\tau}^{\text{EFF}}$ is random due to the presence of CISs.

The following theorem is our main result and characterizes the solution to the optimization problem (2).

THEOREM 1. *There exists $\Lambda \in \mathbb{R}$ such that the optimal threshold $\iota^\star$ for a policy maximizing (2) satisfies*

$$\forall i \in [m] : V(\iota_i^\star; \mathcal{E}_i) = \Lambda \text{ or } (V(\iota_i^\star; \mathcal{E}_i) < \Lambda \text{ and } \iota_i^\star = \infty)$$

*and* $\sum_{i=1}^{m} f(\iota_i^\star, \mathcal{E}_i) = R,$

*where* $V(\iota; \mathcal{E}) = \tilde{\mu} \left( \sum_{i=0}^{\lfloor \frac{t}{\beta} \rfloor} \frac{v^i}{(\Delta + v)^{i+1}} \mathcal{R}_{\exp}^i((\alpha + \gamma)(\iota - i\beta)) \right.$

$$\left. - \sum_{i=0}^{\lfloor \frac{t}{\beta} \rfloor} \frac{\exp(-\alpha\iota)}{\gamma} \mathcal{R}_{\exp}^i(\gamma(\iota - i\beta)) \right)$$

*and* $f(\iota, \mathcal{E}) = \left( \sum_{i=0}^{\lfloor \frac{t}{\beta} \rfloor} \frac{1}{\gamma} \mathcal{R}_{\exp}^i(\gamma(\iota - i\beta)) \right)^{-1}$

*and where $\mathcal{R}_{\exp}^i$ denotes the normalized residual of the $i$-th Taylor approximation of the exponential function $\mathcal{R}_{\exp}^i(x) = \frac{\exp(x) - \sum_{j=0}^{i} \frac{x^j}{j!}}{\exp(x)}$.*

While this does not allow for a simple analytical solution, we can approximate the solution via the following Lemma.

LEMMA 2. *The functions $V$ and $f$ are monotonous in their first argument for any $\mathcal{E}$.*

Lemma 2 suggest that, given oracle access to $V$ and $f$, for any $\Lambda'$ we can find $\iota_i'$ such that $V(\iota_i'; \mathcal{E}_i) \approx \Lambda'$ via line-search and compute its frequency $f(\iota_i'; \mathcal{E}_i)$. We run an outer line search over $\Lambda'$ to find a value of $\Lambda'$ such that $\sum_{i=1}^{m} f(\iota_i'; \mathcal{E}_i) \approx R$.

## 5 A SCALABLE DISCRETE POLICY

Optimal continuous policies are in general undesirable in practice, since the actual bandwidth constraint forbids spikes of crawl events over any time interval, not only asymptotically. This cannot be controlled in the optimal solution of the continuous policy class.

This is why, from an infrastructure point of view, it is desirable to execute a policy from the discrete class.

Azar et al. [1] proposed a transition from their continuous policy to a discrete one via low-discrepancy sampling, a procedure that crucially relies on the fact that the crawl events for each web page under the optimal continuous policy are evenly spaced. Unfortunately, this approach is not practical in a real world crawler for the following two reasons. On the one hand, solving the joint optimization problem, for example the one defined in (5) in Appendix A, is computationally demanding when trillions of pages are in the system. In addition to this, change and request rates are constantly being updated and new pages are added to the pool which requires to again solve the optimization problem to get an update for the crawling policy. On the other hand, in the presence of CI signals, the crawl intervals of the optimal continuous policy are random and depend on the number of CI signals received in any interval. It is unclear how to generalize low-discrepancy sampling to this scenario.

Next we present a more general discretization strategy based on a method presented in *The Unofficial Google Data Science Blog* [10] for the case without the change-indicating signals. Below we extend it to include CISs, as well.

*Discrete policy via bandwidth control.* We propose the following general procedure to derive a discrete policy. The continuous solution induces for any bandwidth $R$ a policy $\pi^R$ that asymptotically satisfies the bandwidth constraint. Instead of running the algorithm with a fixed $R$, we are controlling the bandwidth over time in such a way that the crawl satisfies a discrete policy. At time $t = \frac{j-1}{R}$, pick $R_j$ such that the first crawl of policy $\pi^{R_j}(\mathcal{H}_t)$ happens exactly at time $\frac{j}{R}$. While this sounds like a computationally demanding reduction, this is in fact easier to solve than the continuous problem. We obtain this policy by crawling any $i_t \in \arg\max_{i \in \{1,...,m\}} V(\tau_i^{\text{EFF}}(t); \mathcal{E}_i)$ at any time step. This immediately follows from Theorem 1 by setting the Lagrange multiplier $\Lambda$ to the maximum crawl value.

We note that this policy is fully decentralized expect the arg max operation. This means that it is trivial to incorporate change of parameters and addition or removal of URLs in a fully decentralized manner.

Our empirical evaluations in Section 6 further show that there is small, albeit not significant, improvement in performance compared to the algorithm of [1].

## 5.1 Special cases of the crawl value function

We provide explicit formulas of the crawl value function for different settings.

**No change-indicating signals:** If there are no change-indicating signals available, then this reduces to the vanilla crawl problem studied by Cho and Garcia-Molina [3]. This implies $\tau^{\text{EF}}(t) = \tau^{\text{ELAP}}(t)$, $\alpha = \Delta$ and the value function reduces to $V_{\text{GREEDY}}(\iota; \mathcal{E}) = \frac{\tilde{\mu}}{\Delta} \mathcal{R}_{\exp}^1(\Delta\iota)$.

**Change-indicating signals, no false positives:** Without false positive signals, whenever a signal is provided for a page it implies that the content of a page as outdated. This corresponds to $\beta = \infty$ and $\tau^{\text{EF}}(t) = \tau^{\text{ELAP}}(t)$ if no CI signal has been received and $\tau^{\text{EF}}(t) = \infty$ otherwise. The value function becomes in the limit of these

parameters

$$V_{\text{GREEDY\_CIS}}(\iota; \mathcal{E}) = \begin{cases} \frac{\tilde{\mu}}{\Delta} \text{ if } \iota = \infty \text{ or otherwise} \\ \tilde{\mu} \left( \frac{\mathcal{R}_{\exp}^0((\alpha+\gamma)\iota)}{\alpha+\gamma} - \frac{\mathcal{R}_{\exp}^0(\gamma\iota)}{\gamma \exp(\alpha\iota)} \right) . \end{cases}$$

Notice that $\gamma \to 0$ recovers the value function of without change-indicating signals.

**Noisy change-indicating signals:** In the general case, we obtain the general value function discussed in section 4.

$$V_{\text{GREEDY\_NCIS}}(\iota, \mathcal{E}) = \tilde{\mu} \sum_{i=0}^{\lfloor \frac{t}{\beta} \rfloor} \left( \frac{\nu^i}{(\Delta + \nu)^{i+1}} \mathcal{R}_{\exp}^i((\alpha + \gamma)(\iota - i\beta)) \right.$$
$$\left. - \frac{\exp(-\alpha\iota)}{\gamma} \mathcal{R}_{\exp}^i(\gamma(\iota - i\beta)) \right) .$$

**Approximations of GREEDY_NCIS:** Since the value function in the general case is computationally demanding when $\lfloor \frac{t}{\beta} \rfloor$ is large, we also consider an approximation where we set any higher order residuals to 0 in the computation of the crawl value. We denote the $j$-level approximation of the general value function, summing the first $j$ terms only, by $V_{\text{G\_NCIS-APPROX-J}}$ (the exact formula is presented in Appendix A.1).

The proposed discrete policy computes $V(\tau_i^{\text{EFF}}(t); \mathcal{E}_i)$ for each page independently and picks the page with the largest value, i.e. $\arg\max_{i \in \{1,...,m\}} V(\tau_i^{\text{EFF}}(t); \mathcal{E}_i)$ to crawl at each time step. We call $V(\tau_i^{\text{EFF}}(t); \mathcal{E}_i)$ the crawl value for each page. Thus, only the comparison between the pages with the top crawl values matters in the scheduling result, and the crawl values of most pages do not need to be computed at every time step. We can estimate the crawl value threshold where a page is likely to be selected to crawl by keeping track of the crawl values of the selected pages over time, and estimate the next time when the crawl value of a page needs to be recomputed.

---

**Algorithm 1:** Effective crawling with CI signals

> **Require:** $\forall i : \mathcal{E}_i = (\alpha_i, \beta_i, \gamma_i, \tilde{\mu}_i), R$
>    **for** $t = \frac{1}{R}, \frac{2}{R}, \dots$ **do**
>      Pick Web page $i_t$ to crawl where
>        $i_t \leftarrow \arg\max_{i \in \{1,...,m\}} V(\tau_i^{\text{EFF}}(t); \mathcal{E}_i)$
>      (Instances of $V$ are defined in App. A.1.)
>      Crawl page $i_t$ at time $t$.
>    **end for**

---

## 5.2 Scalable implementation

Other distributed computation technique can also be applied to scale up the computation in practice. For example, we can shard the web pages into $N$ shards and assign $1/N$ bandwidth to each shard, and schedule the web pages in each shard independently in parallel.

## 6 EXPERIMENTS

The goal of our experiments is to support the following claims: (1) Our discrete policy indeed constantly achieves a performance that is close to the optimal continuous one. (2) The CI signals can be

utilized by the proposed methods even if the CI signals are partially observable, come with false positives and are delayed. All results which we report in this study are averaged over 100 repetitions which also allows us to compute standard error for all reported quantities.

## 6.1 Problem instances

A crawling problem when there is no CI signals is fully determined by the change and request rates; we generate these from a uniform distributions (with parameters specified later). The CI signals in our model has three parameters for each page:

**Partially observability parameter ($\lambda_i$):** some part of the change process is not observable in which case the policy does not get CI signals about the change event. The fraction of the change events of page $i$ which is observable by the policy is denoted by $\lambda_i$ and this parameter is generated from a Beta distribution whose parameters are denoted by $\lambda_a$ and $\lambda_b$.

**False positive rate ($v_i$):** the CI signals might contain false positive events in which case not all CIS do correspond to some change event. In our model we assume that the false positive events are also generated according to a Poisson process. The rate with which the false positive events are generated is denoted by $v_i$ for page $i$ and this parameter is generated from a uniform distribution over $[v_{\min}, v_{\max}]$.

## 6.2 Policies

For each problem instance and bandwidth $R$, the optimal policy can be computed by solving (5) which corresponds to the optimal continuous policy, and Algorithm 1 provides our proposed approximation in the space of discrete policies. We consider the performance of the optimal continuous policy as the baseline, and shall refer to this method as BASELINE. The performance of a policy is its accuracy: fraction of events when there is a fresh copy upon request. Note that the accuracy of a continuous policy with rates $(\xi_1, \ldots, \xi_m)$ can be computed as $\frac{1}{\sum_{i=1}^{m} \mu_i} \sum_{i=1}^{m} G(\xi_i; \mu_i, \Delta_i)$.

Our comparison study includes a few policies based on Algorithm 1 using the special cases of value functions presented in Section 5.1. GREEDY uses the value function without knowledge of change-indicating signals. GREEDY-CIS operates under the (possibly false) assumption that change-indicating signals are noiseless. GREEDY-NCIS using the exact value function in the general case and G-NCIS-APPROX-1 and G-NCIS-APPROX-2 are the one or two step approximations of that function.

## 6.3 Accuracy of a policy

Each algorithm was run with a bandwidth $R = 100$ and for $t \leq 1000$, thus each policy schedule $100,000$ crawl events in a single run. We are varying the number of pages $m$ in our experiments to control the hardness of the problem instance at hand. Note that the change parameters $\Delta_i$ as well as the request parameters are drawn from uniform distribution over $[0, 1]$ thus $\sum_{i=1}^{m} \Delta_i$ and $\sum_{i=1}^{m} \mu_i$ are close $m/2$ in expectation. Therefore the expected change and request events given that $t \leq 1000$ are around $(m \cdot 1000)/2$ in a single run with $m$ web pages. We compute the accuracy of a crawling policy over these $(m \cdot 1000)/2$ events, i.e. fraction of request events when

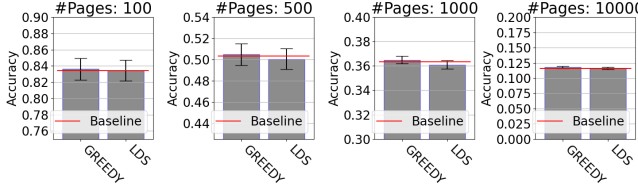

Figure 2: Accuracy of discrete policies without using change-indicating signal. The LDS corresponds to the Algorithm 3 in [1] where the input rates are coming from the solution of (5) with the true change and request rates.

fresh copy is available. Note that the larger the number of pages, the lower accuracy is that is achievable by any policy. We will always report the accuracy of the BASELINE method which corresponds to the optimal continuous policy and can be computed by solving (5).

## 6.4 Continuous vs. discrete policies

Discrete policies have several advantages comparing to continuous ones in a production system. [3] already came up with several simple approaches for converting a continuous policy into a discrete one. In a recent paper, Azar et. al. [1] made use of the technique of low discrepancy sequences [9] which is a scheduling technique for discrete systems so as the empirical rates of each scheduled event has to be close to a predefined rate. Their approach consists of solving the continuous problem, that is given in (5), to obtain optimal rates for the continuous case, and then applying a low discrepancy sequence algorithm to turn this continuous policy into a discrete one. We implement Algorithm 3 of [1] in experiments for comparison, and refer to this approach as LDS.

The GREEDY approach conceptually also converts a continuous policy into a discrete one like LSD algorithm; however, there is no need to solve the continuous problem, but it computes the value of function $V_{\text{GREEDY}}$ for each page which only depends on the elapsed times since the last crawl, and on the change and request rates. Therefore, GREEDY does not require to solve a large constrained optimization before its deployment.

In the first set of experiments, we will compare the performance of GREEDY and LDS. The results are shown in Figure 2. Based on the results, we had found that both policy have a very similar performance regardless of the number of pages and in addition to this, the performance of both algorithm is on par with the performance of the baseline which is the optimal policy in the continuous class. We also compared the empirical crawling frequency of GREEDY and LDS to the frequency of the optimal policy, and we had found that they are close to each other. This analysis is deferred to Appendix B due to space limitations.

## 6.5 Partially observable change sequences

In the second set of experiments, we assess the utility of CI signal when it is partially observable. We generate problem instance as in Subsection 6.4, but this time we compare the performance of GREEDY-CIS to the performance of GREEDY, since GREEDY-CIS is supposed to be amenable to utilize CI signal in an efficient way when there is no false positive CI signal.

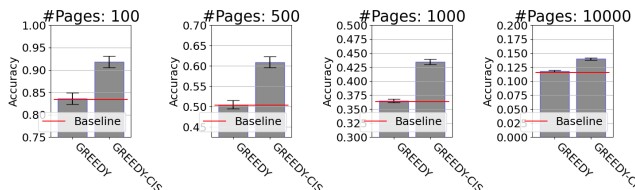

Figure 3: Accuracy of GREEDY and GREEDY-CIS with change-indicating signal.

The parameter $\lambda_i$, which controls the fraction of observable change events according to our model, is generated from Beta$(0.25, 0.25)$ which has a bi-modal density function. Thus, the change events are not revealed too much via CI signals for some pages, and they are revealed almost every time for some other pages. The results are presented in Figure 3. The results clearly show that CI signals can significantly improve the performance of the crawling policy regardless to the relation of bandwidth $(R = 100)$ and sum of change parameters $\sum_{i=1}^{m} \Delta_i$.

Next, we investigate where this performance boost of GREEDY-CIS does come from. For doing so, we plotted the empirical rates achieved by GREEDY and GREED-CIS which are shown in Figure 12. Each dot corresponds to a web page for which we compute the rate of the BASELINE method versus the empirical rates of various methods. The color of the dots indicates the value of parameters $\lambda_i$ which represents the fraction of change rates for which CI signals are available. The larger the parameter $\lambda$, the more CI signals are provided for a web page. Figure 13 shows the very same rates but the color of the dots indicate the change parameter $\Delta_i$ of the web page. Based on the results with 100 web pages that are shown in the top subplots in Figure 13 and 12, one can see that GREEDY-CIS decreases the crawling rate for web pages with many CI signals, and allocates more bandwidth for web pages with no or few CI signal. Interestingly, the results for 300 web pages reveal a different behaviour (see in the bottom subplots of Figure 13 and 12). In this case, some of the web pages with many CIS get boosted and some of them get decreased. Typically, those web pages, which have high change rates, get a higher crawling rate comparing to rate of BASELINE (see bottom right subplot of Figure 13). In general, the rates of those web pages which have no or few CIS, those rates remained close to the optimal rate of the BASELINE method which is the optimal behaviour when no CI signals are provided (see the blue dots around the diagonal of right subplots of Figure 12).

## 6.6 Presence of false positive CIS

In this set of experiments, we compare the performance of various policies when false positive are also present among the change indicating signals, and the policy is aware of the rate of the false positive events. In the first experiment, we run all policies, including GREEDY, GREEDY-CIS, GREEDY-NCIS, G-NCIS-APPROX-1 and G-NCIS-APPROX-2, with $m \in \{100, 200, 500, 750, 1000, 10000\}$ and with a constant bandwidth $R = 100$. Thus, the larger the number of web pages, the less bandwidth can be allocated per web page. The observability parameters $(\lambda_i)$ were generated from Beta$(0.25, 0.25)$, and the parameters that determines the rate of false positive CIS

is generated from Unif$(0.1, 0.6)$. The results are presented in Figure 4. The results revels several trends which we summarize as

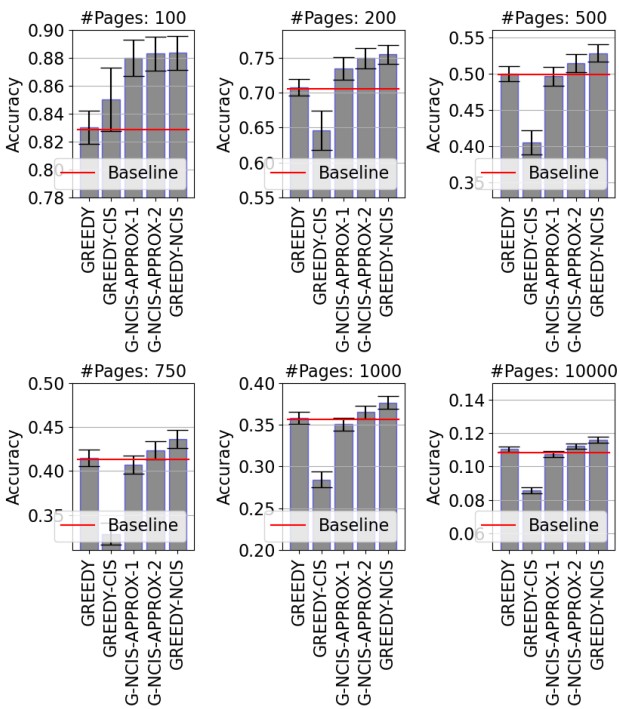

Figure 4: Accuracy for policies with CIS that is partially observable and there are false positives signal. The partially observabilty parameter is generated as $\lambda_i \sim$ Beta$(0.25, 0.25)$, and the rate of false positive CI signals as $v_i \sim$ Unif$(0.1, 0.6)$. The parameter selection is described in Subsection 6.1.

follows. The performance of GREEDY-NCIS, G-NCIS-APPROX-1 and G-NCIS-APPROX-2 is superior to GREEDY and GREEDY-CIS in almost every case, so they can utilize CIS with false positives. When the band width is not tight, i.e. $m \in \{100, 200, 500\}$ the performance of the algorithms with approximated value function, i.e. G-NCIS-APPROX-1 and G-NCIS-APPROX-2) is very close to the performance of the one with exact crawl value, i.e. GREEDY-NCIS. However, for larger number of web pages, the exact computation shows superiority and policies with approximated value function cannot utilize the CI signal. The performance of GREEDY-CIS, which assumes no false positive CI signals, is deteriorated with the number of pages. This policy fully relies on CI signals meaning that it assumes that the content of a page gets outdated upon incoming CI signal. Therefore it allocates more crawl bandwidth to those web pages which have CI signals with many false positive CI signals. This is justified by the empirical rates that can be seen in Figure 14, presented in Appendix F. The results show that GREEDY-CIS achieves significantly higher empirical rates for some pages with high observability rate (indicated by red arrows), whereas GREEDY-NCIS is able to handle this more noisy CI signal efficiently.

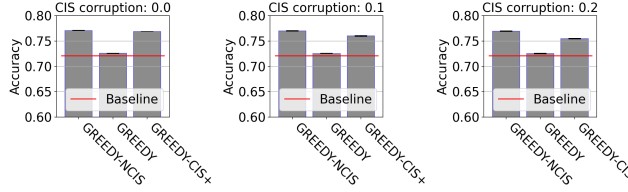

**Figure 5: Accuracy of GREEDY-NCIS,GREEDY and GREEDY-CIS+ with corrupted change-indicating signals on 100k URLs.**

## 6.7 Real world change-rate and precision

Finally, we conduct an experiment with empirical parameters (Figure 5). We take the dataset provided by Kolobov et al. [7] mentioned in Section 2. We follow their protocol to create a semi-synthetic simulation environment: We subsample 100k URLs uniformly from the dataset, and use the provided importance and change-rate information for running simulations with the Poisson model. However, to reflect that CISs are not perfect, we generate CISs differently.

In their paper, Kolobov et al. [7] labelled a set of URLs havin CISs with perfect precision and recall (ca. 5% of sampled URLs). Since our measurements, presented in Section 2, did not validate the presence of such good signals, we apply the following procedure to produce CI signals for our experiments, which both respects the URL selection of [7] and the precision/recall distributions presented in Section 2: We take the fraction of URLs which the dataset labels as perfect precision and recall (ca. 5% of sampled URLs). We split the empirical precision and recall distributions of Section 2 into a lower part consisting of the lowest 95% values and the highest 5% values. We sample precision and recall numbers for the labelled top URLs according to the upper tail of the distribution and for all other URLs from the lower end. We set the crawl budget to 5000 per time step in accordance with prior work, evaluate the policies on 200 time steps and repeat the experiment 10 times.

Since the policy of Kolobov et al. [7] is not trivially extendable to uniform crawling, we use GREEDY-CIS+ for a fair comparison. This algorithm assigns the crawl value of GREEDY for all non high-quality URLs (defined by Precision > 0.7 and Recall > 0.6 to match all top URLs without corruption) and the crawl value of GREEDY-CIS for high quality pages.

To simulate the impact of imperfect estimations, we corrupt all precision and recall numbers by mixing in uniformly sampled noise $\xi_i \sim \text{unif}(0, 1)$ with $precision = (1 - p)precision + p\xi_i, recall = (1 - p)recall + \xi_i$ for $p \in \{0, 0.1, 0.2\}$.

The accuracy of the crawling policies under the different settings are presented in Figure 5. We observe that splitting the pages into high and low quality and using simplified crawl values is close to optimal when the estimations are correct, but is less robust to corrupted precision and recall estimations.

## 7 CONCLUSIONS AND FUTURE WORK

In this paper, we derived several discrete policies which can be implemented easily in a decentralized and scalable way which is demanded by the huge number of URLs that a web caching system has to cope with. In addition, these scalable policies are extended so as they are amenable to handle noisy CI signals with great efficiency. We empirically justified that the performance of the discrete policies is on par with the continuous one that requires lots of centralized computation, and, in addition, they are able to improve their performance with respect to the continuous optimal policy when CI signals are available.

We also tested GREEDY-NCIS when the CI signals are delayed and the bandwidth constraint is changing. In this case, we apply a simple heuristic of discarding CI signals if they arrive shortly after a crawl event (see Appendix C for details). We found that this simple approach can handle delayed CISs if the delays follows an exponential distribution. In addition, we had found that our policies can adjust well to changing bandwidth which result is presented in Appendix D. We plan to devise a more principled way of handling delayed CISs in the future.

There are several research topics that are worth exploring and are related to our work. Maybe the most natural research question to address is how to apply politeness constraint [8] in our setup. Politeness constraints allow to set a bandwidth allocation for a subset of URLs. For example, one can allocate a bandwidth limit to a host. These constraints are more fine-grained than the global bandwidth constraint which is denoted by $R$ in this paper. An another interesting research avenue to explore is how to learn policies in an online fashion when parameters of the system have to be estimated on the fly. This very same question had been addressed by [11] and [6] but without CI signals.

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

# A PROOFS

We provide the proofs for our theoretical results in Section 4.

The following property of the residual function is useful in this section. For ease of notation, from now on we drop the subscript exp for the function $\mathcal{R}_{\exp}$. For any $i > 0$, we have

$$\frac{\partial}{\partial x}\mathcal{R}^i(x) = \frac{\partial}{\partial x}\left[1 - \sum_{j=0}^{i} \frac{x^j}{j!\exp(x)}\right] = \mathcal{R}^{i-1}(x) - \mathcal{R}^i(x). \tag{3}$$

PROOF OF LEMMA 1. Assume the opposite is true and with probability $p > 0$, a crawl event is not triggered by a threshold rule. This implies than we can find consecutive crawl intervals such that $\tau^{\mathrm{EF}}(t_i^{\mathrm{CR}}) > \tau^{\mathrm{EF}}(t_{i+1}^{\mathrm{CR}})$, such that there is no CI signal directly at $t_i^{\mathrm{CR}}$ and that these crawl intervals have positive volume on the real line. We show that one can increase the objective by moving the crawl event, which means that the current policy was not optimal. Pick a small time $\epsilon$, such that no CI signal falls between $[t_i^{\mathrm{CR}} - \epsilon, t_i^{\mathrm{CR}}]$ and assume that the crawl event would have happened at time $t_i^{\mathrm{CR}} - \epsilon$ instead. The original unnormalized objective over the two crawl intervals is

$$\int_{t_{i-1}^{\mathrm{CR}}}^{t_i^{\mathrm{CR}}} \exp(-\alpha\tau^{\mathrm{EF}}(s))\,ds + \int_{t_i^{\mathrm{CR}}}^{t_{i+1}^{\mathrm{CR}}} \exp(-\alpha\tau^{\mathrm{EF}}(s))\,ds.$$

After changing the policy by shifting the value, we have an objective of

$$\int_{t_{i-1}^{\mathrm{CR}}}^{t_i^{\mathrm{CR}}-\epsilon} \exp(-\alpha\tau^{\mathrm{EF}}(s))\,ds + \int_0^\epsilon \exp(-\alpha s)\,ds + \int_{t_i^{\mathrm{CR}}}^{t_{i+1}^{\mathrm{CR}}} \exp(-\alpha(\tau^{\mathrm{EF}}(s)+\epsilon))\,ds.$$

The difference in objective is

$$(1 - p_i \cdot \exp(\alpha\epsilon))\int_0^\epsilon \exp(-\alpha s))\,ds - (1 - \exp(-\alpha\epsilon)) \times$$

$$\int_{t_i^{\mathrm{CR}}}^{t_{i+1}^{\mathrm{CR}}} \exp(-\alpha\tau^{\mathrm{EF}}(s))\,ds \geq \frac{1 - \exp(-\alpha\epsilon)}{\alpha}(p_{i+1} - p_i \cdot \exp(\alpha\epsilon)),$$

where the last inequality follows from

$$\int_{t_i^{\mathrm{CR}}}^{t_{i+1}^{\mathrm{CR}}} \exp(-\alpha\tau^{\mathrm{EF}}(s)) \leq \int_0^{\tau^{\mathrm{EF}}(t_{i+1}^{\mathrm{CR}})} \exp(-\alpha s)\,ds = \frac{1 - p_{i+1}}{\alpha}.$$

Since We assumed $p_i < p_{i+1}$, we can find an $\epsilon > 0$ such that the objective increases with our policy change. Hence the original policy was not optimal.                    □

PROOF OF LEMMA 3. We can consider $t_1^{\mathrm{CR}}$ as a function of the threshold $\iota$ and consider any realization of the CI signal process at which changing the threshold changes the first crawl event. By the Leibniz integral rule for differentiation, we have

$$\frac{\partial}{\partial t}\int_0^{t_1^{\mathrm{CR}}(\iota)} \exp(-\alpha\tau^{\mathrm{EF}}(s))\,ds = \exp(-\alpha\tau^{\mathrm{EF}}(t_1^{\mathrm{CR}}(\iota)))\frac{\partial}{\partial t}t_1^{\mathrm{CR}}(\iota).$$

We assume changing the threshold changes the time of crawl, hence $\tau^{\mathrm{EF}}(t_1^{\mathrm{CR}}(\iota)) = \iota$. Taking the expectation over all these trajectories finishes the proof.                    □

PROOF OF LEMMA 2. Using Lemma 3 and

$$V(\iota) = \tilde{\mu}(w(\iota) - \exp(-\alpha\iota)\psi(\iota))$$

yields

$$V'(\iota) = \tilde{\mu}\alpha\exp(-\alpha\iota)\psi(\iota) > 0.$$

Hence the function $V$ is monotonically increasing. The derivative of the function $\mathcal{R}^i(x)$ is

$$\mathcal{R}^{i-1}(x) - \mathcal{R}^i(x) = \frac{x^i}{i!}\exp(-x) > 0.$$

Hence the derivative of the function $\psi(\iota)$ is strictly positive and the function $f$ is monotonically decreasing.                    □

PROOF OF THEOREM 1. The contribution of page $i$ to the overall objective $O(\pi(\iota))$ (i.e., its weighted expected freshness) can be expressed as

$$o(\iota_i; \mathcal{E}_i) = \tilde{\mu}_i \mathbb{E}_{\pi(\iota), t\sim\mathrm{unif}(\mathbb{R}_+)}\left[\exp(-\alpha_i\tau_i^{\mathrm{EFF}}(t))\right],$$

which again only depends on the parameters $\iota_i$ and $\mathcal{E}_i$. Then

$$O(\pi) = \sum_{i=1}^{m} o(\iota_i; \mathcal{E}_i),$$

and finding the optimal policy with bandwidth constraint $R$ that maximizes (2) reduces to the optimization problem

$$\text{maximize} \quad \sum_{i=1}^{m} o(\iota_i; \mathcal{E}_i) \quad \text{subject to} \quad \sum_{i=1}^{m} f(\iota_i; \mathcal{E}_i) \le R,$$

or by substituting $\xi_i = f(\iota_i; \mathcal{E}_i)$, we have equivalently

$$\text{maximize} \quad \sum_{i=1}^{m} o(f^{-1}(\xi_i; \mathcal{E}_i); \mathcal{E}_i) \quad \text{subject to} \quad \sum_{i=1}^{m} \xi_i \le R, \tag{4}$$

where the inverse of $f$ is with respect to its first argument. Since the crawl intervals $t_{i,n+1}^{\mathrm{CR}} - t_{i,n}^{\mathrm{CR}}$ are i.i.d. random variables for any $n \in \mathbb{N}$ due to the Poisson processes, we can rewrite the frequency as the inverse of the average length between two subsequent crawls.

The objective in (4) is the sum of the expected freshness of the different web pages at a random point in time, weighted by the request intensity $\tilde{\mu}_i$, similarly to the seminal work of [1]. In fact, it is easy to show that (4) matches the objective in [1] in the absence of CI signals, in which case $\alpha_i = \Delta_i$ for all $i \in [m]$, and the crawling interval is deterministic with length $\iota_i$. Consequently, picking a random point in time is equivalent to picking a random point in any interval between two crawls, since all intervals are of the same length. So in this case, for any threshold $\iota \ge 0$ and Poisson parameters $\mathcal{E}$, the objective function boils down to

$$o(\iota; \mathcal{E}) = \tilde{\mu} \cdot \frac{1}{\iota} \int_0^{\iota} \exp(-\Delta s) \, ds = \tilde{\mu} \cdot \frac{1}{\Delta \iota}(1 - \exp(-\Delta \iota)),$$

and

$$o(f^{-1}(\xi; \mathcal{E}); \mathcal{E}) = G(\xi; \tilde{\mu}, \Delta) := \frac{\tilde{\mu}}{\Delta} \xi \left(1 - \exp\left(-\frac{\Delta}{\xi}\right)\right).$$

In the absence of CISs, the optimal policy is given by the optimization problem

$$\text{maximize} \quad \sum_{i=1}^{m} G(\xi_i; \mu_i, \Delta_i) \quad \text{subject to} \quad \sum_{i=1}^{m} \xi_i \le R. \tag{5}$$

which is a special case of (4). The solution $\xi^* = (xi_1^*, \dots, \xi_m^*)$ of (5) results in a policy which crawls web page $i$ in optimal intervals of length exactly $1/\xi_i^*$. (In fact, this optimization objective is already introduced in equation 6 of [1], where it is referenced from [3].)

Before we derive the functions $o$ and $f$ for the web-ping setting, we first discuss how to actually solve the optimization problem (4) in the general form. By the Karush–Kuhn–Tucker conditions [2], any local optimum $\xi^*$ satisfies for all $i \in [m]$

$$\frac{\partial}{\partial x} o(f^{-1}(x; \mathcal{E}_i); \mathcal{E}_i)\Big|_{x=\xi_i^\star} = \Lambda$$

$$\text{or} \; \frac{\partial}{\partial x} o(f^{-1}(x; \mathcal{E}_i); \mathcal{E}_i)\Big|_{x=\xi_i^\star} < \Lambda \text{ and } \xi_i^* = 0$$

for some Lagrange multiplier $\Lambda \ge 0$ (obtained by solving (4) with Lagrange's method). The case $\xi_i^* = 0$ corresponds to never crawling a web page. This can be the optimal for maximizing the objective if there are web pages with low importance $\tilde{\mu}$ and high change rate $\Delta$. In practice, completely abandoning web pages might be unacceptable and can be alleviated by enforcing a hard threshold $\xi_i > \varepsilon$ on the crawl frequency.

Under sufficient regularities, e.g., $o \circ f^{-1}$ being concave, this is also a sufficient condition for optimality. Define the function

$$V(\iota; \mathcal{E}) = \frac{\partial}{\partial x} o(f^{-1}(x; \mathcal{E}); \mathcal{E})\Big|_{x=f(\iota; \mathcal{E})},$$

then any optimal threshold $\iota^\star$ satisfies

$$V(\iota_i^\star; \mathcal{E}_i) = \Lambda \qquad \text{or} \qquad V(\iota_i^\star; \mathcal{E}_i) < \Lambda \text{ and } \iota_i^\star = \infty$$

subject to the bandwidth constraint $\sum_{i=1}^{m} f(\iota_i^\star, \mathcal{E}_i) = R$.

*Computing $V$ and $f$ with CI signals.* We define the following auxiliary functions for a policy parameterized by $\iota$ so as we drop the web page subscript of all quantities, since we are considering a single web page:

$$w(\iota; \mathcal{E}) = \mathbb{E}_{\pi(\iota)}\left[\int_0^{t^{\mathrm{CR}}} \exp(-\alpha \tau^{\mathrm{EFF}}(s)) \, ds\right] \tag{6}$$

$$\psi(\iota; \mathcal{E}) = \mathbb{E}_{\pi(\iota)}\left[t^{\mathrm{CR}}\right]. \tag{7}$$

These quantities are the expected cumulative freshness between two consecutive crawl events and the expected length of that interval, respectively. These two functions are closely related as the following lemma shows.

LEMMA 3. *The derivatives of $w, \psi$ with respect to the first argument satisfy for any environment $\mathcal{E}$ $w'(x; \mathcal{E}) = \exp(-\alpha x)\psi'(x; \mathcal{E})$.*

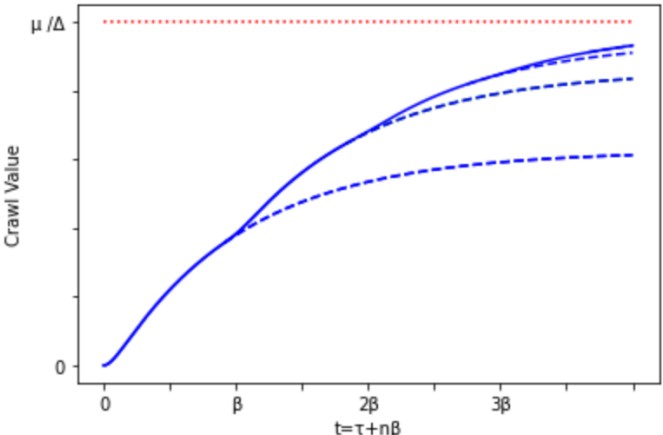

**Figure 6: Example of the crawl-value function $V$. Dashed lines are approximations when terminating the sum after 1, 2 or 3 steps respectively. Red line is the asymptotic value of all functions for $\iota \to \infty$.**

These two functions allow us to express $V$ and $f$ nicely. As mentioned before, the frequency is simply the inverse expected length, so $f(\iota; \mathcal{E}) = 1/\psi(\iota; \mathcal{E})$. The objective $o$ is given by

$$o(\iota; \mathcal{E}) = \tilde{\mu} \cdot w(\iota; \mathcal{E}) \cdot f(\iota; \mathcal{E}),$$

hence by the inverse function rule, we have

$$V(\iota; \mathcal{E}) = \tilde{\mu} \left. \frac{\partial}{\partial x} \left[ w(f^{-1}(x; \mathcal{E}); \mathcal{E}) \cdot x \right] \right|_{x = f(\iota; \mathcal{E})}$$

$$= \tilde{\mu} \left( w(\iota; \mathcal{E}) + \frac{w'(\iota; \mathcal{E})}{f'(\iota; \mathcal{E})} \right)$$

$$= \tilde{\mu} \left( w(\iota; \mathcal{E}) - \exp(-\alpha \iota) \psi(\iota; \mathcal{E}) \right),$$

where the last equation uses Lemma 3 and $\frac{1}{f'(x)} = -\frac{\psi(x)}{\psi'(x)}$. Finally, we present the analytical solutions for the functions $\psi$ and $w$.

LEMMA 4. *The expected length of an interval between two consecutive crawl events and its cumulative freshness under the threshold-policy $\pi(\iota)$ in environment $\mathcal{E}$ are given by*

$$\psi(\iota; \mathcal{E}) = \sum_{i=0}^{\lfloor \frac{t}{\beta} \rfloor} \frac{1}{\gamma} \mathcal{R}_{\exp}^i (\gamma(\iota - i\beta))$$

$$w(\iota; \mathcal{E}) = \sum_{i=0}^{\lfloor \frac{t}{\beta} \rfloor} \frac{\nu^i}{(\Delta + \nu)^{i+1}} \mathcal{R}_{\exp}^i ((\alpha + \gamma)(\iota - i\beta)),$$

*where $\mathcal{R}_{\exp}^i$ denotes the normalized residual of the $i$-th Taylor approximation of the exponential function $\mathcal{R}_{\exp}^i(x) = \frac{\exp(x) - \sum_{j=0}^i \frac{x^j}{j!}}{\exp(x)}$.*

Note that the computational complexity of evaluating the functions $\psi$, $w$ grows in $t$. To avoid unbounded computation, one can approximate the function values very well by terminating the summation after a small finite number of steps (e.g. 3, see Figure 6), since the residual of the $i$-th Taylor approximation converges quickly to 0.                                                                                                                                                                                     □

Proof of Lemma 4. We begin with the function $\psi$. Assume $\iota \in [0, \beta]$. Then the length of the first interval is given by $\min\{t_1^{\text{CIS}}, \iota\}$, since the first received CI signal puts the effective time above the threshold. The expected length is given by

$$\psi(\iota; \mathcal{E}) = \int_0^\iota \gamma \exp(-\gamma s) s \, ds + \exp(-\gamma \iota) \iota$$

$$= [-\exp(-\gamma s) s]_0^\iota - \int_0^\iota -\exp(-\gamma s) \, ds + \exp(-\gamma \iota) \iota$$

$$= \frac{1 - \exp(-\gamma \iota)}{\gamma}$$

$$= \frac{\mathcal{R}^0(\gamma \iota)}{\gamma},$$

which is the same function value as claimed. Now by induction, the show that the function is correct for any $\iota$. Assume that the formula is correct for any $\iota' \in [0, i\beta]$, then we show that it is also correct for $\iota \in (i\beta, (i + 1)\beta]$. The expected length between crawls can be given recursively by the time that passes until $t_1^{\text{CIS}} \in [0, \iota - i\beta]$ plus the length for the remaining threshold, or $\iota - i\beta$ plus the length for a threshold $i\beta$ if no CI signal was received in $[0, \iota - i\beta]$ (omitting $\mathcal{E}$ for brevity)

$$\psi(\iota) = \mathbb{E}\big[\mathbb{I}\{t_1^{\text{CIS}} \in [0, \iota - i\beta]\}(t_1^{\text{CIS}} + \psi(\iota - \beta - t_1^{\text{CIS}})) + \mathbb{I}\{t_1^{\text{CIS}} > \beta\}(\iota - i\beta + \psi(i\beta))\big]$$

$$= \int_0^{\iota - i\beta} \gamma \exp(-\gamma s)(s + \psi(\iota - \beta - s)) \, ds + \exp(-\gamma(\iota - i\beta))(\iota - i\beta + \psi(i\beta))$$

$$= \psi(\iota - i\beta) + \int_0^{\iota - i\beta} \sum_{j=0}^{i-1} \frac{\mathcal{R}^j(\gamma(\iota - (j + 1)\beta - s))}{\exp(\gamma s)} \, ds + \exp(-\gamma(\iota - i\beta))\psi(i\beta).$$

The middle term is

$$\int_0^{\iota - i\beta} \sum_{j=0}^{i-1} \frac{\mathcal{R}^j(\gamma(\iota - (j + 1)\beta - s))}{\exp(\gamma s)} \, ds$$

$$= \sum_{j=0}^{i-1} \left[ -\frac{\mathcal{R}^{j+1}(\gamma(\iota - (j + 1)\beta - s))}{\gamma \exp(\gamma s)} \right]_0^{\iota - i\beta}$$

$$= \sum_{j=0}^{\lfloor \frac{\iota}{\beta} \rfloor} \frac{\mathcal{R}^j((\iota - \beta j)\gamma)}{\gamma} - \frac{\mathcal{R}^0(\gamma \tau)}{\gamma} + \frac{\mathcal{R}^0(\gamma i\beta)}{\gamma \exp(\gamma(\iota - i\beta))} - \exp(-\gamma(\iota - i\beta))\psi(i\beta)$$

$$= \sum_{j=0}^{\lfloor \frac{\iota}{\beta} \rfloor} \frac{\mathcal{R}^j((\iota - \beta j)\gamma)}{\gamma} - \psi(\iota - i\beta) - \exp(-\gamma(\iota - i\beta))\psi(i\beta).$$

Combining this with the previous equation finishes the derivation for $\psi(\iota)$.

Next, we show that the claimed function for $w$ is correct. Note that $w(0) = 0$ which is the correct value. By Lemma 3, it is sufficient to show that the ratio of derivatives between $\psi(x)$ and $w(x)$ matches. Recall

$$\frac{\partial}{\partial x} \mathcal{R}^i(x) = \mathcal{R}^{i-1}(x) - \mathcal{R}^i(x) = \frac{x^i}{i!} \exp(-x).$$

Hence

$$\frac{\partial}{\partial x} \psi(x)\bigg|_{x=\iota} = \sum_{i=0}^{\lfloor \frac{\iota}{\beta} \rfloor} \frac{(\gamma(\iota - i\beta))^i}{i! \exp(\gamma(\iota - i\beta))}.$$

The derivative of our claimed form of $w$ is

$$\frac{\partial}{\partial x} w(x)\bigg|_{x=\iota} = \sum_{i=0}^{\lfloor \frac{\iota}{\beta} \rfloor} \frac{\gamma^i}{(\alpha + \gamma)^i} \frac{((\alpha + \gamma)(\iota - i\beta))^i}{i! \exp(\alpha\iota + \gamma(\iota - i\beta))}$$

$$= \sum_{i=0}^{\lfloor \frac{\iota}{\beta} \rfloor} \frac{(\gamma(\iota - i\beta))^i}{i! \exp(\gamma(\iota - i\beta))} \exp(-\alpha\iota).$$

Hence the ratio of the derivatives satisfy Lemma 3, which implies that we have found the correct analytical form of $w(x)$. □

## A.1 Crawl value based on approximation

**Approximations of Greedy_NCIS:** Since the value function in the general case is computationally demanding when $\lfloor \frac{t}{\beta} \rfloor$ is large, we also consider an approximation where we set any higher order residuals to 0 in the computation of the crawl value. We denote the $j$-level approximation of the general value function by

$$V_{\text{G\_NCIS-Approx-J}}(\iota, \mathcal{E}) = \tilde{\mu} \sum_{i=0}^{\min\{j-1, \lfloor \frac{\iota}{\beta} \rfloor\}} \left( \frac{v^i}{(\Delta + v)^{i+1}} \mathcal{R}^i_{\exp}((\alpha + \gamma)(\iota - i\beta)) - \frac{\exp(-\alpha\iota)}{\gamma} \mathcal{R}^i_{\exp}(\gamma(\iota - i\beta)) \right).$$

## B COMPARISON OF EMPIRICAL RATES OF GREEDY AND LDS

Even if the performances of the GREEDY and LDS algorithms are very similar, they implement quite different scheduling strategies. When we compare the empirical rates of GREEDY and LDS to the rates of the Baseline method that can be obtained by solving (5), we can see that even if the performances of these two algorithms are on-par, the rates for the same pages are different from each other. Figure 7 shows the empirical rates achieved by GREEDY and LDS with 100 and 500 web pages takes from 10 problem instances. The empirical rates of LDS are very close to the optimal continuous policy rates since dots are on the diagonal red line. This result can be explained by the fact that the LDS is based on a low discrepancy scheduling algorithm whose objective is to directly minimize the deviation of these two rates over time. However, the GREEDY takes into account the staleness of web pages when it picks the next web page to be crawled.

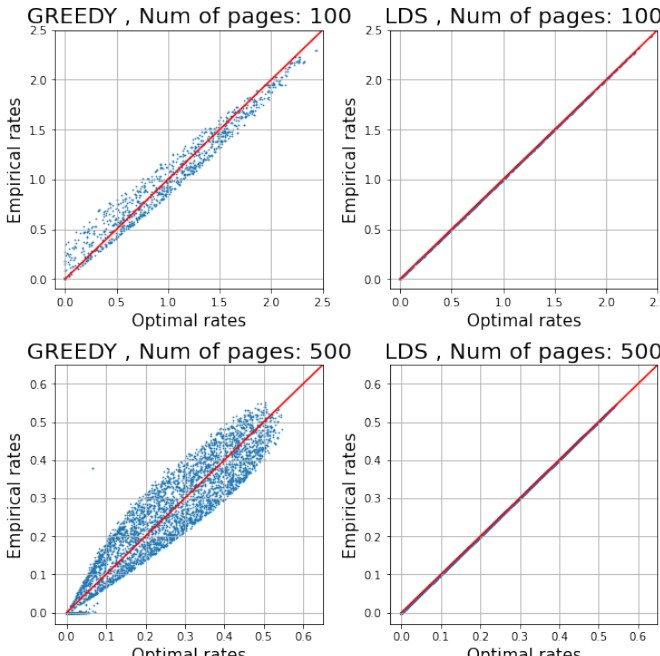

**Figure 7: Empirical rates for various web pages achieved by discrete policies without using change-indicating signals. Each dot corresponds to a web page for which the optimal rate versus the empirical rate of the corresponding method is plotted. The web pages are taken from 10 synthetic problem instances with 100 and 500 web pages, respectively. The optimal rates are computed based on (5), and they correspond to the Baseline method. LDS corresponds to Algorithm 3 of [1] with the rates of Baseline method as input.).**

## C DELAYED CI SIGNALS

So far, we have assumed that CI signals arrive instantaneous with the corresponding change event. In practice, there is some latency either due to the network or the entity generating the CI signals itself. However, the bulk of web pages changes on a scale of several days, and for such pages any CI signal delay is minuscule compared to the average interval between change events. We propose to simply discard CI signals that arrive within an interval $[t^{\text{CR}}, t^{\text{CR}} + T_{\text{DELAY}}]$ after a crawl for some tuneable parameter $T_{\text{DELAY}}$, to ensure that we do not make decisions based on out-dated CI signals.

Next experiment, we assess the impact of the delay of CI signals on the performance of GREEDY-NCIS policy and to what extent the proposed thresholded approach, defined in Section C can remedy the performance drop that is caused by the delay of CI signals. The

thresholding consists of discarding the CI signal for a page $i$ when it is close to a recent crawl event that fetches the content of page $i$. We set this time window to $T_{\text{DELAY}} = 5/R$.

The problem instances we use in this experiment are generated in a similar way as in the previous section: the partially observabilty parameter is generated as $\lambda_i \sim \text{Beta}(0.25, 0.25)$, and the rate of false positive CI signals is $v_i \sim \text{Unif}(0.1, 0.6)$. In addition to this, the CI signals are delayed with a random quantity that is generated from the Poisson distribution with $v = 6$ as it is described in Section 6.1.

Figure 8 shows the result with the delayed CI signals. We indicate the baseline by the red line as before which is the performance of the optimal continuous policy without using CI signals. In addition to this we also indicate the performance of GREEDY-NCIS policy without delayed CI signals by the blue line. The performance that is indicated by the blue line coincides with the one reported in Figure 4. Based on the results, one can see that the delay of the CI signals indeed deteriorates the performance of GREEDY-NCIS when the bandwidth is not so tight, and has marginal impact when the crawling problem has tighter bandwidth allocation. This can be explained by the fact that when the bandwidth is tighter the improvement achieved by GREEDY-NCIS over GREEDY policy, which does not use CI signals, is smaller, which implies that the CI signals do not help so much in this case. Thus, their delay has not so significant impact in that case, either. Nevertheless, when we compare the performance of GREEDY-NCIS-D to the performance of GREEDY-NCIS with no delay (blue line), we see that when $m = 100$, their performances are on-par. In this case, this simple thresholding policy can almost recover the performance drop caused by delay.

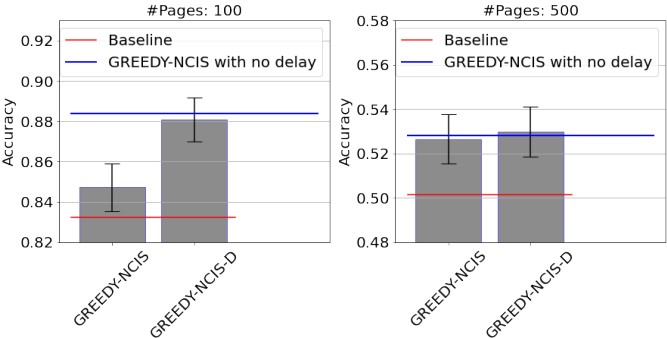

Figure 8: Accuracy achieved by discrete policies including GREEDY-NCIS and GREEDY-NCIS-D. The partially observabilty parameter is generated as $\lambda_i \sim \text{Beta}(0.25, 0.25)$, and the rate of false positive CI signals as $v_i \sim \text{Unif}(0.1, 0.6)$. In addition, the CI signals are delayed with a random quantity that is generated from the Poisson distribution with $v = 6$. The parameter selection is described in Subsection 6.1.

## D    BURN-IN TIME

In the next experiment, we demonstrate that the discrete policy GREEDY is able to automatically adjust the prioritization of web pages to new optimal solutions when the total bandwidth changes, without centralized computation. In this example, the total bandwidth starts from $R = 100$, and suddenly changes to $R = 150$, before changing back to $R = 100$, with $m = 1000$ web pages to crawl, running for $t \leq 400$. The GREEDY policy is used to select web pages, and there is no extra computation when the total bandwidth changes. Figure 9 shows that the accuracy automatically rises to the new optimal level when the bandwidth increases, and automatically falls back to the original optimal level when the bandwidth decreases back. The accuracy which is plotted in Figure Figure 9 is computed on the last 1000 crawl events for every time step.

## E    ESTIMATING MODEL PARAMETERS

We directly observe all request events and CI signals, hence we can estimate the importance parameters $\mu$ and the rate of the CIS $\gamma$ with good accuracy based on the overall frequency of logged events. The unobserved change rate $\alpha$ and "goodness" of CI signals $\beta$ are harder to estimate, since only partial knowledge available of the content change events.

A naive approach is to use empirical crawl events which are observed to create an approximation of the change sequence and directly calculate the empirical precision and recall of the CI signals based on this approximation. To evaluate this approach, we conduct the following experiment: We sample precision and recall values uniformly between $[0.2, 0.95]$. The expected lenth of the change process is sampled uniformly between $[2, 20]$ and the relative rate of crawls is set between four times to one quarter of the change rate. We use the models to

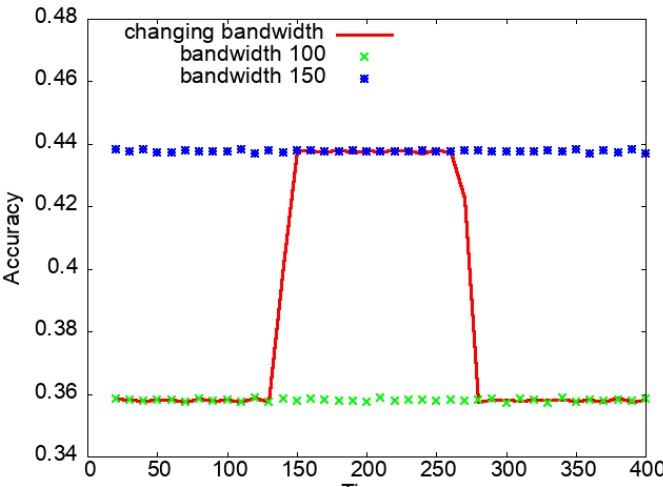

**Figure 9: Accuracy of GREEDY over time. The red line shows how the accuracy of GREEDY changes over time, when the total bandwidth starts from 100, and increases to 150 at time 133, before decreasing back to 100 at time 266. The green line shows the accuracy of GREEDY, when the total bandwidth is always 100. The blue line shows the accuracy of GREEDY, when the total bandwidth is always 150.**

create synthetic datasets of time horizon 100000 and reconstruct precision and recall based on either a statistical approach that computes

$$\text{precision} = \frac{\text{number of intervals with CI signal and change}}{\text{number of intervals with change}}$$

$$\text{recall} = \frac{\text{number of intervals with CI signal and change}}{\text{number of intervals with change}},$$

or fitting the linear model for $\alpha$ and $\beta$ to the data and compute precision and recall based on these.

Figure 10 shows that the statistical estimator is clearly biased.

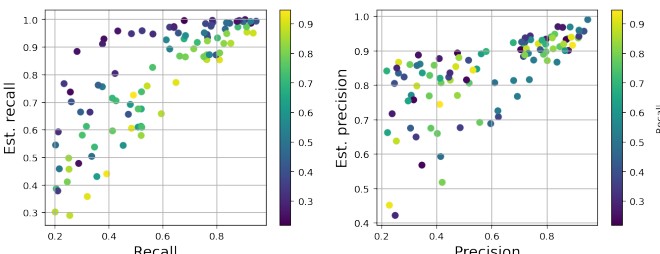

**Figure 10: Bias of the naive estimator of precision and recall of CIS.**

We achieve more faithful results by fitting a model to the empirical data. We collect the data $\left(\left(\binom{\tau^{\text{ELAP}}}{n^{\text{CIS}}}_1, z_1\right), \dots\right)$, where $z_i$ is a binary variable indicating whether there has been a change between crawl $i - 1$ and $i$. The model predicts that $z_i \sim \text{Ber}(\exp(-\langle\binom{\alpha}{\alpha\beta}, \binom{\tau^{\text{ELAP}}}{n^{\text{CIS}}}_i\rangle))$. Estimating the unknown parameter vector $\binom{\alpha}{\alpha\beta}$ can be done via MLE. In our experiments, we assume that these parameters are known to the policies, but in a production system this estimation can be carried out based on logged data. The absolute error of this estimator was on the order of $10^{-4}$ as shown in Figure ?? .

## F  EMPIRICAL RATES WITH AND WITHOUT FALSE POSITIVES

We compared the empirical rates of various policies. This is presented in 14. The results show that GREEDY-CIS achieves significantly higher empirical rates for some pages with high observability rate (indicated by red arrows), whereas GREEDY-NCIS is able to handle this more noisy CI signal efficiently. This means that in the presence of false positive signals, GREEDY-CIS which assumes no false positive CIS, overly relies on the side information, i.e. the false positives boosts its empirical crawling rate unnecessarily.

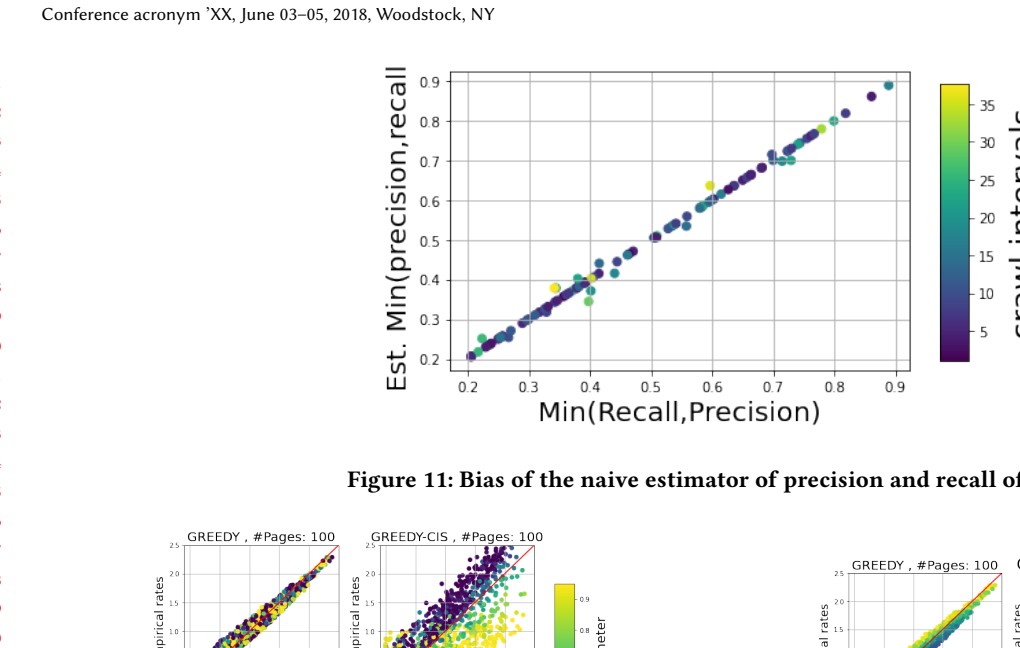

Figure 11: Bias of the naive estimator of precision and recall of CIS.

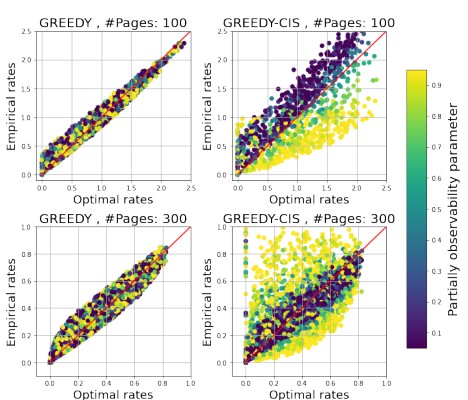

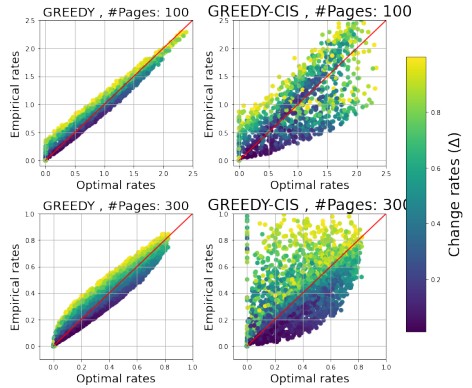

Figure 12: Empirical rates for various web pages achieved by discrete policies including GREEDY and GREEDY-CIS. Each dot corresponds to a web page for which we computed the rate of the BASELINE method and plotted it versus the empirical rate achieved by the corresponding policy. The color of the dots indicates the observability of the change sequence that is controlled by $\lambda$.

Figure 13: Similar to Figure 12 but the color of dots indicates the change rates of the web pages.

Received 20 February 2007; revised 12 March 2009; accepted 5 June 2009

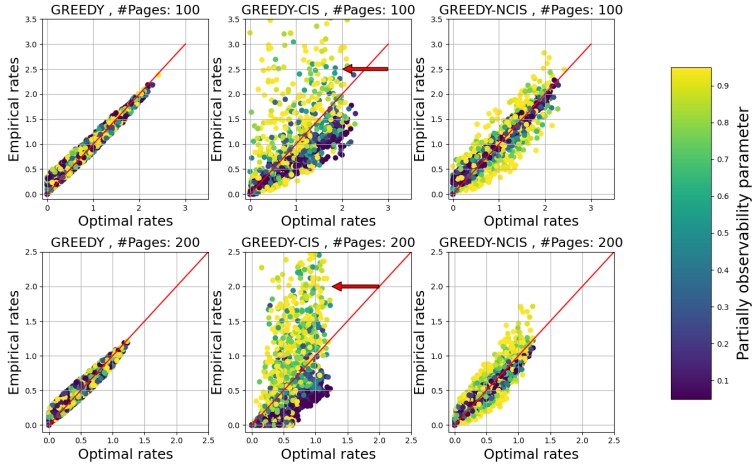

**Figure 14: Empirical rates achieved for various web pages achieved by discrete policies including GREEDY, GREEDY-CIS and GREEDY-NCIS. Each dot corresponds to a web page for which we computed the rate of the BASELINE method and plotted it versus the empirical rate achieved by the corresponding policy. The partially observabilty parameter is generated as $\lambda_i \sim$ Beta$(0.25, 0.25)$, and the rate of false positive CI signals as $v_i \sim$ Unif$(0.1, 0.6)$. The parameter selection is described in Subsection 6.1.**

