# OpenReview forum: "A Scalable Crawling Algorithm Utilizing Noisy Change-Indicating Signals"
_ACM.org/TheWebConf/2025/Conference — WWW 2025 Poster_

### Official Review · Reviewer_7pC5 · 2024-11-19

**Novelty:** 6
**Technical Quality:** 5

**Review:**

The authors build upon previous research on refreshing crawled web pages. Unlike prior work, they assume that change-indicating signals (CISs) are available to provide information on when a web page may have changed. These signals can be noisy and may have false positives. The authors expand the crawling problem to account for the noisy CISs and derive an optimal continuous crawling policy. Additionally, they developed a practical discrete crawling policy based on this method. The authors conducted experiments using synthetic and semi-synthetic data to demonstrate the effectiveness of their approach. The results show that the performance of the discrete policies is comparable to that of the continuous policy, and further improvements are observed when CISs are available.

The task addressed in the paper is interesting, and the proposed approach appears to be novel. However, the paper is somewhat dense and challenging to read. The authors attempted to include a lot of information within the allotted space, negatively impacting the paper’s readability. Notably, Figures 12-14 are discussed in the main text but are presented in the appendix, which may seem unfair to other authors who have tried to include the necessary information in the main text, reserving only supplementary material for the appendix. The authors might consider a different presentation style, such as converting some figures into tables, to save space and ensure all essential information is included in the main text.

Additionally, I would appreciate further discussion of the results. Specifically, in Figure 5, there is almost no change in accuracy across different CIS corruption rates. Can the authors explain why this is the case? Figures 2-4 show a substantial drop in accuracy as the number of pages increases. A discussion on why this decline is so steep would be beneficial. Furthermore, I would like to get some clarification regarding the differences between G-NCIS-APPROX-1 and G-NCIS-APPROX-2, as the latter often outperforms the former.

**Questions:**

See the above review.

**Reviewer Confidence:**

2: The reviewer is willing to defend the evaluation, but it is likely that the reviewer did not understand parts of the paper

**Scope:**

3: The work is somewhat relevant to the Web and to the track, and is of narrow interest to a sub-community

---

### Official Review · Reviewer_peJ8 · 2024-11-20

**Novelty:** 4
**Technical Quality:** 4

**Review:**

This paper proposes discrete crawling policies that could be implemeted in a scalable manner, addressing the challenge of handling the massive number of URLs in web caching systems. These policies are extended to efficiently manage noisy signals. Experimental results demonstrate that the performance of the discrete policies is comparable to that of the continuous policies.

However, there is some lack of clarity. For example, in Lines 106–142, the discussion focuses on the disadvantages of previous methods, but the explanation is somewhat disorganized and could benefit from better structure.

Pros：
- The problem is of good practical value and the motivation is clearly stated.
- The proposed crawling policy can deal with imperfect CI signals with great efficiency.

Cons:
The paper writing is not good. The paper could be further polished to make it structure concise and centered around the topic 'scalable'. Some sentences are hard to follow, such as Line 186 "In this section, we present some to basic statistics about change
detection signals in general and in the dataset we use in our experiments. "

**Questions:**

The algorithm claims to be scalable, but the current organization in Section 5 does not clearly demonstrate this. Does “scalable” here refer to the ability to perform parallel computation? If so, this aspect is not explicitly addressed or validated in the experimental section.

**Reviewer Confidence:**

2: The reviewer is willing to defend the evaluation, but it is likely that the reviewer did not understand parts of the paper

**Scope:**

4: The work is relevant to the Web and to the track, and is of broad interest to the community

---

### Official Review · Reviewer_rSNs · 2024-11-22

**Novelty:** 4
**Technical Quality:** 4

**Review:**

**Summary**
- This research proposes a scalable web crawling framework that incorporates noisy Change-Indicating Signals (CIS) to optimize freshness under bandwidth constraints. The authors introduce Greedy-NCIS, a policy that dynamically adjusts crawling decisions based on the quality of signals (precision and recall) while balancing freshness and resource efficiency. They validate the method using semi-synthetic datasets, demonstrating that it outperforms baseline approaches in handling noisy and incomplete signals. The work highlights the importance of integrating signal reliability into crawling policies and offers scalable solutions for large-scale web systems.

**Strengths**
- The paper is well-motivated by its focus on evaluating the quality of existing Change-Indicating Signals (CIS), which effectively highlights the need to account for noisy CIS.
- The introduction of approximation-based policies (NCIS-APPROX) to balance performance and scalability is both innovative and pragmatic.
- Additionally, the experiments are thoughtfully designed to demonstrate the effectiveness of incorporating NCIS across datasets of varying scales, adding depth to the study’s practical relevance.

**Weaknesses**
- Related Works
  - The absence of a Related Work section makes it difficult to understand prior research and see how the proposed method distinguishes itself in terms of novelty.
- Experiments
  - The paper lacks sufficient comparison with state-of-the-art approaches for previous crawling methods..
  - The reliance on semi-synthetic data for experiments raises concerns about the generalizability of the findings.The dataset combines statistically generated data (Poisson processes for change and request rates) with real-world-inspired parameters (CIS precision and recall) to simulate realistic crawling scenarios, but its real-world applicability remains uncertain.
- Others
  - Additionally, some mathematical notations and algorithmic steps are not clearly explained.

**Questions:**

- The authors appear to use [1] as the main baseline for their work. However, it is challenging to discern the specific features of [1] and how the proposed method differs from it besides using change-indicating signals.
- In line 208-215, the authors note that [2] uses its own definition of ‘change’ and they have similarly chosen to define it in their own way. Is there a specific rationale for deviating from the definition provided by [2]?
- How does the complexity of Greedy-NCIS scale for a dataset with millions of pages, such as in the real-world?
- Could the authors provide more details on how the thresholds for Greedy-NCIS influence the performance in practice?

[1] AZAR, Yossi, et al. Tractable near-optimal policies for crawling. Proceedings of the National Academy of Sciences, 2018, 115.32: 8099-8103.

[2] KOLOBOV, Andrey, et al. Staying up to date with online content changes using reinforcement learning for scheduling. Advances in Neural Information Processing Systems, 2019, 32.

**Reviewer Confidence:**

3: The reviewer is confident but not certain that the evaluation is correct

**Scope:**

3: The work is somewhat relevant to the Web and to the track, and is of narrow interest to a sub-community

---

### Official Review · Reviewer_mSVS · 2024-11-22

**Novelty:** 6
**Technical Quality:** 6

**Review:**

𝐒𝐮𝐦𝐦𝐚𝐫𝐲:
This paper addresses the problem of optimizing web crawling to maintain the freshness of cached web pages while utilizing noisy change-indicating signals (CIS). The authors propose a scalable crawling algorithm that integrates noisy CIS, adapts to bandwidth constraints, and supports decentralized implementation. The paper outlines both continuous and discrete crawling policies and demonstrates the algorithm's effectiveness through theoretical analysis and empirical experiments on semi-synthetic data. The results show that the proposed discrete policies can achieve comparable performance to continuous ones while being more practical for large-scale web crawling systems.

𝐒𝐭𝐫𝐞𝐧𝐠𝐭𝐡𝐬:
- 𝐒𝟏: The paper tackles a critical challenge in web crawling: balancing the freshness of cached pages with limited bandwidth, a problem that has significant implications for search engine performance and environmental impact.

- 𝐒𝟐: The paper provides a thorough theoretical foundation for incorporating noisy CIS into crawling policies, deriving optimal strategies for both continuous and discrete settings, and presenting clear algorithms for implementation.

- 𝐒𝟑: The proposed discrete crawling algorithm is highly scalable and can be implemented in a decentralized manner, which is crucial for handling trillions of web pages.

- 𝐒𝟒: The authors conduct extensive experiments to validate the algorithm’s performance under various conditions, including noisy and partially observable CIS, showcasing its robustness and versatility.

𝐖𝐞𝐚𝐤𝐧𝐞𝐬𝐬𝐞𝐬:
- 𝐖𝟏: While the paper presents semi-synthetic experiments, it lacks real-world deployment scenarios to validate the algorithm’s performance in live web crawling systems. This raises questions about its practicality in uncontrolled environments.

- 𝐖𝟐: The computational demands of the algorithm, especially for large-scale implementations, are not fully addressed. While the decentralization aspect is highlighted, the complexity of computing crawl values for noisy CIS remains a concern.

- 𝐖𝟑: The paper does not explicitly address ethical issues such as respecting website owners' preferences, managing the impact of crawling on server loads, or mitigating potential misuse of CIS data.

𝐒𝐮𝐠𝐠𝐞𝐬𝐭𝐢𝐨𝐧𝐬 𝐟𝐨𝐫 𝐈𝐦𝐩𝐫𝐨𝐯𝐞𝐦𝐞𝐧𝐭:
- Use visual aids (e.g., diagrams, flowcharts) to explain complex algorithmic details and improve readability for a broader audience.

- Add a dedicated section discussing ethical implications, including privacy, server load management, and the fairness of resource allocation among web pages.

- Provide a detailed discussion on the computational complexity of the algorithm, particularly for large-scale systems, and propose strategies to mitigate overhead.

**Questions:**

- How does the algorithm handle dynamic changes in CIS noise patterns over time? Can it adapt without significant retraining or recomputation?

- How do false positives in CIS affect the fairness of crawling resource allocation across web pages? Are there safeguards to prevent over-crawling certain sites?

**Reviewer Confidence:**

4: The reviewer is certain that the evaluation is correct and very familiar with the relevant literature

**Scope:**

4: The work is relevant to the Web and to the track, and is of broad interest to the community